# Individual and Contextual Determinants of Flu Vaccination Adherence: A University Nudge Intervention

**DOI:** 10.3390/ijerph20105900

**Published:** 2023-05-21

**Authors:** Nadia Pecoraro, Francesca Malatesta, Luna Carpinelli, Domenico Fornino, Claudio Giordano, Giuseppina Moccia, Matilde Perillo, Mario Capunzo, Giulia Savarese, Francesco De Caro

**Affiliations:** 1Department of Medicine, Surgery and Dentistry—Scuola Medica Salernitana, University of Salerno, 84081 Baronissi, Italy; 2Hospital “San Giovanni di Dio e Ruggi d’Aragona”, 84081 Salerno, Italy

**Keywords:** flu vaccine, confidence, nudge intervention

## Abstract

Introduction: The University of Salerno has implemented a nudge intervention with the aim of promoting vaccine adherence among employees of academia and identifying individual and contextual determinants that influence adherence. Method: A purpose-built questionnaire was used during the reference period of October–December 2022 in order to assess levels of state anxiety (STAI-Y1), perceived stress (PSS-10), and public sentiments, which influence vaccination behavior, with consequences for the whole population (VCI). Results: Analysis of the results revealed a difference in mean scores on the PSS: those who have always adhered to the vaccination campaign compared to those who have never been vaccinated perceived higher levels of stress (12.01 vs. 11.33; F = 4.744, *p* = 0.031); furthermore, there was a relationship between the presence/absence of pathologies and VCI (F = 3,93; df = 1; *p* = 0.04). Conclusions: The University of Salerno’s nudge intervention made its employees more responsible for protecting the health of the academic community and encouraged good adherence to the flu vaccination campaign. University employees, equipped with high cultural tools, sought information primarily from institutional sources indicated by the university during the free vaccination campaign at the university’s vaccine center.

## 1. Introduction

Flu is a significant public health challenge, with up to 50 million symptomatic cases and from 15,000 to 70,000 flu-related deaths occurring each year in Europe according to the European Centre for Disease Prevention and Control (ECDC). Vaccination is the most effective method of preventing seasonal flu, and the World Health Organization (WHO) has adopted a strategy to combat flu with the objective of increasing vaccination coverage, as stated in the Global Flu Strategy (2019–2030). This strategy is particularly important in the current context as the simultaneous circulation of SARS-CoV-2, respiratory syncytial virus (Rsv), and flu virus, even in mild or moderate forms, could put pressure on the healthcare system due to case management and disease complications.

Non-pharmaceutical interventions (NPIs), such as mask use, frequent hand washing, reduced social contacts, and social distancing, have been implemented since March 2020 to contain SARS-CoV-2. These measures have also reduced the spread of other pathogens, including flu. In 2020 and 2021, the global circulation of the seasonal flu virus was limited due to these interventions [1]. However, Italy’s 2022–2023 season has seen a higher peak incidence value of flu-like syndromes than in previous years [2]. It has been hypothesized that the use of personal protective equipment (PPE) may have caused a lack of immune stimulation, resulting in “immunity debt,” which impairs the immune system’s ability to maintain itself in “training” and increases susceptibility to the virus following re-exposure [3]. Frequent stimulation of the innate immune system through vaccinations or infections can improve the defensive response to infection [4].

In recent decades, a growing number of individuals have become more critical of recommended vaccinations, necessitating further research into the determinants of vaccination adherence. Vaccination adherence is a behavior resulting from a decision-making process that may be influenced by various factors [5,6]. The Strategic Advisory Group of Experts (SAGE) on Immunization established the SAGE Working Group on Vaccine Hesitancy in March 2012, defining “vaccine hesitancy” as a delay in accepting or refusing vaccination despite its availability [5,7]. Vaccine hesitancy represents a significant global health threat as it undermines effective and cost-effective vaccination programs. The SAGE Model of determinants of vaccine hesitancy identifies three key domains: contextual influences, individual and social group influences, and specific issues related to the vaccine or vaccination process [5,7]. Hesitancy ranges from complete acceptance to complete refusal of the vaccine, and those who fall within the continuum may refuse some vaccines but accept others (hesitant conformists) [8,9]. Vaccination decision-making also includes individuals who adhere to recommended vaccinations immediately, as they view vaccination as a shared norm, and those who take time to consider the pros and cons by seeking information from family, friends, or community members, searching the Internet, or asking their doctor for advice [10]. SAGE has emphasized the importance of activating institutional and organizational systems locally and globally to promote vaccination adherence [11]. Vaccination adherence encompasses a set of behaviors performed by individuals upon the completion of vaccination, and individuals may trust vaccines but not view a specific vaccination as necessary.

Numerous individual and contextual factors impact adherence to vaccination. Individual factors comprise education level [5], knowledge of infection and its consequences, risk perception [12], information-seeking behavior [13], health and prevention beliefs, and psychological traits such as anxiety levels [14] and stress. Contextual factors consist of socioeconomic status, cultural background, religion, geographic barriers, access to vaccinations (such as organizational management, on-site vaccinations, and free vaccine supply), social approval, and communication and media (including information campaigns).

Confidence in vaccination is a significant factor associated with adherence to vaccines. It is defined as confidence in the vaccine (as a product), the healthcare professional (as a provider), and the decision-makers (as policymakers) [8].

The “3 Cs” model identifies several factors that hinder trust in vaccines and the willingness to undergo vaccination: confidence (in the vaccine, the institutions offering it, the institutes producing it, and doctors and health personnel; influenced by political and religious ideologies), convenience (referring to accessibility to vaccines such as clinic hours, remoteness of clinics, and user-borne costs), and complacency (referring to perceptions of risk for vaccine-preventable diseases and the importance of immunization) [15].

The Vaccine Confidence Index (VCI) [8,16] measures vaccination confidence and is used to monitor and map trust in vaccines globally. The VCI measures trust across four dimensions: confidence in the importance of vaccines, their safety, their efficacy, and the compatibility of vaccines with religious or personal beliefs [17,18]. Conversely, a lack of trust can be influenced by negative experiences, distrust in political decision makers, or particular religious or philosophical beliefs [8].

Instead, Betsch et al. in the “5C Scale” [17,19] measure psychological antecedents that influence vaccination behavior: confidence (in the efficacy and safety of vaccines, but also in the health care system and the professionals who administer them), complacency (when risk perception with respect to the disease is low and vaccination is not considered necessary), constraints (such as the physical availability of vaccines, accessibility to services, ability to understand the problem), calculation (the extent to which a person researches information and conducts their own individual risk-benefit analysis), and collective responsibility (the willingness of people to protect others by vaccinating). This tool makes it possible to classify the psychological antecedents of vaccination and facilitate the design and evaluation of vaccination interventions. In the literature, we have not identified nudge experiences to favor vaccination campaigns in university contexts. This fact has led us to implement and experiment with an intervention at our university.

### 1.1. University of Salerno’s Nudge Intervention

Nudging is considered one of the most effective techniques for behavior modification. It involves using gentle “nudges” that influence the architecture of choice without using persuasion or financial incentives while respecting freedom of choice [20]. Previous studies [21] have shown that nudge interventions tend to be well accepted if the “nudge” is transparent and can be actively managed, thus triggering thought processes. Nudge interventions can take place through different modalities, such as reminders and recalls, different ways of accessing information, message delivery procedures, and the use of emotional associations (e.g., videos and pictures) [22]. Using nudging, a person’s behavior can be predictably modified toward an end. Therefore, the nudge approach can also be effective in promoting vaccine adherence, as advocated by the World Health Organization (WHO) in 2019 [23] in the field of healthcare.

Locally, the University of Salerno, in collaboration with its Department of Medicine, Surgery and Dentistry, in agreement with the Territorial Health Service and with the support of the School of Specialization in Hygiene and Preventive Medicine of the University of Salerno, has implemented a nudge intervention with the aim of promoting a “Flu Vaccination Campaign for the University’s Teaching and Technical-Administrative Staff” in 2021–2022 and then in 2022–2023. The vaccination campaigns were carried out at the outpatient clinics of the university’s competent doctor, where staff go for scheduled medical examinations.

In 2021, the University of Salerno had already engaged in the COVID-19 vaccination of its staff at the Vaccine Center at the University Hospital “San Giovanni di Dio e Ruggi d’Aragona” (Salerno, Italy).

### 1.2. Aims

The aim of this study was to identify individual and contextual determinants that influence confidence in flu vaccination in order to create a strategic organizational model to encourage adherence through a nudge intervention.

## 2. Materials and Methods

### 2.1. Procedure and Data Collection

The survey involved University of Salerno staff who were contacted via institutional email, signed by the University Rector, and informed about the Flu Vaccine Campaign for staff, with the dates and times of the clinics held by the attending physician at the University. Staff could make reservations through a form available in an attached link. The Flucelvax Tetra flu vaccine was free for all participants. The survey was conducted using a purpose-created questionnaire consisting of four sections and was entered on the Google Forms platform via a QR code during the reference period from October to December 2022. Before completing the survey, participants were informed of the research’s purpose and provided informed consent about the use of their data in an anonymous and aggregate form, in accordance with the General Data Protection Regulation of the European Union. The survey was designed to take approximately 20 min to complete.

### 2.2. Instruments

The questionnaire was structured to assess individual data, including demographic information, education level, presence of pathologies, previous flu and COVID-19 vaccinations, sources of information about flu vaccines, and health information levels of state anxiety (State-Trait Anxiety Inventory Y Form—STAI-Y1) [24], perceived stress (Perceived Stress Scale 10—PSS-10) [25], and public sentiments that influence vaccination behavior and have consequences for the whole population (Vaccine Confidence Index-VCI) [8]. The following standardized scales were used to assess stress and anxiety states and the VCI index:-The State-Trait Anxiety Inventory (STAI-Y) is divided into two scales, each consisting of 20 items, which assess state (Y1) and trait anxiety (Y2), respectively. The subject indicates, on a 4-point Likert scale (1 = not at all; 4 = very much), how well the different statements fit their behavior. In the present study, only the Y1 scale was used, which investigates transitory psychological and physiological reactions directly related to adverse situations at a specific time, such as the COVID-19 pandemic.-The PSS-10 measures self-reported stress and was used because of its established validity and reliability. It includes 10 questions, with answers ranked using a 5-point Likert scale and assesses stressful experiences and responses to stress over the previous 4 weeks. Questions that relate to negative events or responses are scored in a reverse manner. The scores range from 0 to 56, with higher scores indicating higher levels of perceived stress.-The Vaccine Confidence Index (VCI) considers eight Likert-type statements included in the staff questionnaire to which participants were asked to declare their agreement or disagreement. The statements were as follows: A1: Flu is a serious illness. A2: Flu vaccine is effective. A3: Healthcare workers must get vaccinated. A4: By getting vaccinated, I protect people close to me from flu. B1: It is better to get flu than the vaccination. B2: Flu vaccines have serious side effects. B3: The vaccine can cause flu. B4: Opposed to vaccination. The level of agreement or disagreement was scored as follows: “totally agree” = 4, “partially agree” = 3, “partially disagree” = 2, and “totally disagree” = 1. For the first four statements (A1–A4), the higher the Likert score, the better the propensity towards vaccines, while for the other four (B1–B4), the higher the Likert score, the lower the propensity. The VCI was calculated as follows: VCI = [(A1 + A2 + A3 + A4)/4]/[(B1 + B2 + B3 + B4)/4] (1), where A1, A2, A3, and A4 were the scores to the first four statements, while B1, B2, B3, and B4 were the scores of the other four statements.

### 2.3. Participants

A total of 250 University of Salerno employees have joined the anti-flu vaccination campaign; of this, 17% of the University’s teaching and administrative employee population characterized our sample.

One hundred sixty-five employees of the University of Salerno (F = 38%; M = 62%; mean age = 52.75, SD = 8.3) took part in the survey, voluntarily and anonymously.

In relation to the type of work role in the university, 43% were administrative employees, 54% were teachers, 0.6% were freelance professionals, and 2.4% had other types of jobs.

With regard to marital status, 63.6% of the participants were married, 13.4% were separated/divorced, 17.6% were single, 3% were widowed, and 2.4% were cohabiting. Among the participants, 83.6% were cohabiting with family or roommates, while 16.4% lived alone.

The level of schooling was as follows: 12.1% had a secondary school degree, 27.9% had a university degree, 0.6% had a lower secondary school degree, and 59.4% had post-graduate training.

Regarding current health status, 44% of the participants had no diseases, while 56% had diseases, including chronic heart disease, hypertension, diabetes mellitus, chronic lung disease, immunosuppressive therapies, oncological diseases, and hematopoietic stem cell transplantation (see Table 1).

Regarding contact with the flu, 94% of the participants had had the flu, while 6% had never had the flu. Of the people who had had the flu, 17% had it last year, 30.4% in the last 2–3 years (including the time of the pandemic), 14.5% four years ago, 1.8% every year, 2.4% never, and 33.9% did not remember.

With respect to the flu vaccine, 5.5% had never been vaccinated, while 40% were vaccinated last year and 54.5% in previous years.

Regarding promoting the vaccine, 21.2% did not invite anyone to get vaccinated, while the remaining 78.8% referred their family members or friends to get vaccinated.

Despite having been vaccinated in previous years, 19.6% of the participants got sick, while 80.4% did not.

Regarding COVID-19, 60.5% of the participants had gotten infections, while 39.5% did not. Among the participants, 3% had two booster doses, 89.1% had three booster doses, and 7.9% had four booster doses (see Table 2).

The participants learned about the vaccination campaign using a variety of sources: 40% used media such as radio and TV, 42.4% used the web and social media, 10.3% learned through face-to-face communication, and 7.3% learned through doctors.

The type of approach to information that is conveyed by professional culture is very important (Chi-square = 31.45; df = 12; *p* = 0.002): workers used media such as TV or radio, as well as medical consultation; university teachers used sources from the web.

Regarding satisfaction with government vaccination campaigns, 63.6% of participants were satisfied, 16.4% were not satisfied, 18.2% had a neutral position, and 1.8% had not considered vaccination campaigns (see Table 3).

## 3. Results

The IBM SPSS v.28 software (IBM^®^ SPSS^®^, Milan, Italy) was utilized for the descriptive analysis of the variables under investigation, as well as for the comparison between the means of the scores obtained from the administered tests.

### 3.1. Psychological States

#### 3.1.1. Anxiety

The participants’ stress levels were within the range of “mild form” (mean = 44.63; SD = 3.75). The teaching staff displayed significantly higher levels of anxiety compared to administrative employees (F = 3.70; df = 3; *p* = 0.013), and there was also a significant relationship with educational attainment (F = 3.077; df = 3; *p* = 0.029) in terms of low (middle school) vs. high (postgraduate education). Those who, despite having the vaccine in past years, got sick, showed higher levels of anxiety (F = 5.94; df = 1; *p* = 0.016).

#### 3.1.2. Stress

Stress levels were in the “low” range (mean = 11.98; SD = 5.98). Stress correlated negatively with age (R = −0.167; *p* = 0.032) and anxiety (R = −0.187; *p* = 0.016). The analysis of the results showed a difference in mean scores on the PSS: those who had always adhered to the vaccination campaign had higher levels of perceived stress than those who had never been vaccinated (12.01 vs. 11.33; F = 4.744, *p* = 0.031). Among those with medical conditions at the time of administration, younger people appeared to be more stressed (F = 5.613; df = 1; *p* = 0.021).

### 3.2. VCI Index

The VCI test score was X = 1.174, SD = 0.50, with a value placed between the 50th and 75th percentiles. There was a significant difference between those who had diseases and those who did not (mean = 1.261; SD = 0.602 vs. mean = 1.106; SD = 0.396) at the time of vaccine administration (F = 3.933; df = 1; *p* = 0.04). Profession also appeared to be a determinant (F = 2. 868; df = 3; *p* = 0.049), with higher vaccine confidence in administrative employees than in teachers (mean = 1.301; SD = 0.599 vs. mean = 1.072; SD = 0.37). There was also a significant relationship with educational attainment (F = 2.989; df = 3; *p* = 0.033).

Those who had never had the flu had a significantly higher confidence index (mean = 1.68; SD = 1.00 vs. mean = 1.14; SD = 0.438; F = 2.74; df = 1; *p* = 0.001), and confidence was distributed differently depending on when the flu appeared in past years (F = 3.34; df = 5; *p* = 0.007), with a higher mean for those who had never had the flu (mean = 2.045; SD = 0.228) or who had the flu every year (mean = 1.308; SD = 0.44) (see Table 4).

In teachers, the VCI was influenced (F = 5.709; df = 2; *p* = 0.005) by the sources of vaccine used (Beta = 0.303; t = 2.85; *p* = 0.005) and the degree of satisfaction in the vaccination campaign (Beta = 0.274; t = 2.58; *p* = 0.001).

## 4. Discussion

There were two aims of this study: (1) to identify individual and contextual determinants that influence confidence in flu vaccination and (2) to create a strategic organizational model to encourage adherence through a nudge intervention.

Results for aim 1: identify individual and contextual variables that influence vaccine confidence (VCI). The contextual variables we considered included a culture of belonging, which is understood about professional roles, communication, and media (information campaigns). Our findings indicate that confidence in vaccination is influenced by the chosen information sources and the proposed vaccination campaigns. Specifically, we found that teachers were more likely to refer to web and social media sources, in addition to traditional media, which significantly influenced their satisfaction with the vaccination campaign and their confidence in the vaccine. On the other hand, administrative employees were more influenced by sources such as television, radio, and doctor’s support, and were more confident in the vaccine. However, administrative employees have a medium-high level of education, which was statistically different from teachers who have post-graduate training. Communication was not a determining factor, but poor or inadequate communication can negatively influence vaccine adherence and contribute to vaccine hesitancy [5]. We hypothesize that access to more critical sources, such as in the case of teachers, may lead to a more thoughtful and less spontaneous vaccine choice. Both professional groups, however, received the same form of communication for vaccination.

Regarding individual variables, we considered socio-demographic variables, health status, perceived risk, previous history of flu and SARS-CoV-2 vaccination, as well as state anxiety [26] and perceived stress [27]. Our results suggest that the most confident individuals are those who have not had the flu or experience the flu every year. Vaccination confidence was high in those who get the flu every year and prepare for the virus through preventive strategies. Those who have not had the flu showed equal confidence, this suggests that there is a perception of increased risk because of the elimination of protective measures, such as masks, and the propensity for vaccine adherence.

Vaccine adherence could also be related to factors typical of the corporate nudge intervention [28] such as vaccine availability, trust in the institution, or collective responsibility [16,17,18,19]. Regarding collective responsibility, it is important to highlight that 78.8% of university employees referred their friends or family members to vaccination; this figure suggests the building of a pro-vaccine psychological architecture [19]. It could also be hypothesized that it is the employees’ attitude of collective responsibility, evidenced in their eagerness to get their relatives and friends vaccinated, that was instrumental in their adherence to the university vaccine program.

The presence of other diseases as a risk factor significantly predisposed individuals to rely on vaccines. Anxiety does not have a direct effect on vaccination confidence, but it was higher in those who have been vaccinated in recent years but still fell ill. Stress also did not directly affect vaccination confidence, but it was present in a mild form in participants and was more prevalent among younger people and those who have previously participated in vaccination campaigns.

Results for aim 2: how to create a strategic organizational model to encourage adherence through a nudge intervention in the university context.

University Rectors could create a specific support structure with psychologists and medical professionals and this staff could:-Investigate the individual and contextual determinants that influence confidence in flu vaccination (online questionnaire; focus groups; individual interviews) [11,26];-Investigate the university employees’ cognitive processes and behaviors to achieve well-suited and educationally sound and responsible in the university context;-Manage a Type 2 nudge, rather than engage the automatic system in the university employees, but do this to trigger reflective thinking that subsequently shapes behavior. Type 2 nudges can create persistent behavioral change, using psychological mechanisms such as memory of past utility, self-perception, and repetition. For example, the support staff could ask their university employees to promise to be on time. This commitment nudge could initially support punctuality, but then, via the paths to persistence, become a new habit of the university employees, even if the initial promise has been forgotten;-Apply a transparent nudge provided in such a way that the intention behind it, as well as how behavioral change is pursued, could reasonably be expected to be transparent to the university employees being nudged as a result of the intervention, for example, using self-persuasion [29];-Focus on the effect of descriptive social norms on desired behaviors that university employees may engage in at suboptimal levels. Specifically, university employees could be more likely to get a flu shot and advocate vaccination when if they know that the majority of their colleagues got vaccinated against seasonal influenza compared to when most colleagues do not;-Implement policy to predicably alter high-stakes behaviors among university employees through low-powered incentives [30];-All university employees could receive a reminder mailing that lists the times and locations of the relevant vaccination spaces. Mailings to employees randomly assigned to the treatment conditions additionally could include a prompt to write down either (i) the date the employee planned to be vaccinated or (ii) the date and time the employee planned to be vaccinated [31].

## 5. Conclusions

As the COVID-19 pandemic subsided, the incidence of flu infections began to rise. To address this, the World Health Organization (WHO) has called for structured prevention interventions, including vaccination campaigns at both national and local levels. To this end, the University of Salerno initiated a nudge intervention for their vaccination campaign, starting with the SARS-CoV-2 vaccine and later the flu vaccine. This intervention aimed to build a pro-vaccine psychological architecture by implementing actions that create psychological antecedents that influence vaccination behavior, as suggested by the 5C model [19].

The University of Salerno has established trust with the healthcare and organizational system, including the ASL (Local Health Authority), by welcoming university employees to the company medical officer’s clinic for vaccination. The participants have built trust with healthcare professionals through previous experiences, such as medical examinations conducted by the company medical officer. Risk perception, measured through the presence of other diseases, was found to be a significant factor in vaccine confidence.

One of the main elements of the nudge intervention was the availability and accessibility of free vaccines without the need to visit one’s primary care doctor. The current nudge intervention informed university employees that the vaccine is a community protection tool, both in the work organization and in the family context, activating the cognition of collective responsibility. It would be useful to investigate how and where the cognitive construct of collective responsibility was constructed: whether this caring was transferred from the private life context to work, fueling the vaccine choice, or whether conversely, the promotion of the vaccine at work then raised awareness among employees who referred their family members.

Information about the vaccination campaign was provided both through email and during vaccine administration at the outpatient clinic. However, the study was limited in that it did not include a control sample of unvaccinated individuals, and it would be useful to extend the study longitudinally to determine the long-term effect of the nudge intervention on vaccine confidence and adherence. This would help to identify which aspects of the nudge intervention were effective in promoting vaccine confidence and adherence.

## Figures and Tables

**Table 1 ijerph-20-05900-t001:** Percentage of frequencies (%) relating to the descriptive variables.

Descriptive Variables	Response Set	Frequencies (%)
Gender	Men	62%
	Women	38%
Type of work role in the university	Administrative employees	43%
	Teachers	54%
	Freelance professionals	0.6%
	Other	2.4%
Marital status	Married	63.6%
	Separated/divorced	13.4%
	Single	17.6%
	Widowed	3.0%
	Cohabiting	2.4%
Live	Cohabiting with family or roommates	83.6%
	Live alone	16.4%
Level of schooling	Secondary school degree	12.1%
	University degree	27.9%
	Lower secondary school degree	0.6%
	Post-graduate training	59.4%
Health status of university employees	Administrative employees	44%
	Teachers	56%

**Table 2 ijerph-20-05900-t002:** Percentage of frequencies (%) relating to contact with flu, COVID-19, and vaccines.

	Response Set	Frequencies (%)
Contact with flu	Yes	94%
	No	6%
When was contact with flu	Never	2.4%
	Last year	17%
	Last 2–3 years	30.4%
	4 years ago	14.5%
	Every year	1.8%
	Do not remember	33.9%
Flu vaccination	Never vaccinated	5.5%
	Vaccinated last year	40%
	Vaccinated previous years	54.5%
Promoting vaccine	Did not invite anyone	21.2%
	Invite their family members/friends	78.8%
Despite previous flu vaccine, they have had flu in previous years	Yes	19.6%
	No	80.4%
Sick with COVID-19	Yes	60.5%
	No	39.5%
COVID-19 vaccine doses administered	2 booster doses	3%
	3 booster doses	89.1%
	4 booster doses	7.9%

**Table 3 ijerph-20-05900-t003:** Percentage of frequencies (%) relating to information about the flu vaccine.

	Response Set	Frequencies (%)
Source about campaign vaccination	Media (radio/TV)	40%
	Press	42.4%
	Face-to-face communication	10.3%
	Doctors	7.3%
Satisfaction with vaccination campaigns	Satisfied	63.6%
	Not satisfied	16.4%
	Neutral position	18.2%
	Had not considered vaccination campaigns	1.8%

**Table 4 ijerph-20-05900-t004:** Values of the VCI index in the distribution of the variables.

VCI Index	Response Set	Mean	SD
Professional role	Administrative employees	1.301	0.599
	Teachers	1.072	0.37
Other diseases at time of vaccine administration	Yes	1.261	0.602
	No	1.106	0.396
Educational attainment	Secondary school degree	1.360	0.778
	University degree	1.292	0.567
	Post-graduate training	1.083	0.369
Flu experience	Yes	1.142	0.438
	No	1.68	1.0
When was contact with flu	Never	2.045	0.228
	Last year	1.191	0.570
	Last 2–3 years	1.115	0.550
	4 years ago	1.294	0.530
	Every year	1.308	0.444
	Do not remember	1.174	0.352

## Data Availability

Written informed consent was obtained from the subjects in order to publish this paper.

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
