# Peer review of "Individual and Contextual Determinants of Flu Vaccination Adherence: A University Nudge Intervention"

_ijerph, 2023, doi:10.3390/ijerph20105900_

Round 1

Reviewer 1 Report

The paper covers a relevant and up-to-date issues which makes it worthy to be published. I only have a few suggestions that I hope will improve its readability.  The section titled “University of Salerno's nudge intervention” needs to be adjusted. The title suggests a more detailed description of the University of Salerno intervention, instead most of the section is dedicated to a general description of what the Nudge intervention is. Please, make text and title more coherent. Furthermore, at the end of this section the authors report the aim. This should be in a dedicated section and should be a bit more elaborated.

The aim is the following: “identify individual and contextual determinants that influence confidence in flu vaccination in order to create a strategic-organizational model to encourage adherence through a nudge intervention”. My feeling is that the final discussion is limited to the first part of the aims (identify individual and contextual determinants). The second part, about the strategic-organizational model, is quite neglected. I would suggest either to elaborate more on this in the Conclusions or not to include this part in the aims.

Please specify what ASL stays for.

Author Response

The authors are grateful for the careful reading and the precious comments received. The suggestions formulated by the reviewers have been highly supportive in the improvement of the manuscript and deepening of several key aspects.

Comment: The paper covers a relevant and up-to-date issues which makes it worthy to be published. I only have a few suggestions that I hope will improve its readability.  The section titled “University of Salerno's nudge intervention” needs to be adjusted. The title suggests a more detailed description of the University of Salerno intervention, instead most of the section is dedicated to a general description of what the Nudge intervention is. Please, make text and title more coherent.

Answer: The authors express their gratitude to the referee for his constructive comments. The manuscript has been revised and improved following the reviewer suggestions. The added/modified text has been highlighted in yellow in the revised manuscript. The authors agree with the referee about the slight mismatch between text and title and revised the manuscript accordingly. The updated Discussion section covers, on one hand, the identification of the main determinants affecting the confidence towards the flu vaccination in the investigated population and, on the other hand, the definition of a strategic-organizational model promoting the adherence through a nudge intervention.

Comment: Furthermore, at the end of this section the authors report the aim. This should be in a dedicated section and should be a bit more elaborated.

Answer: With the purpose of evidencing the motivations for the implementation of the described work, a dedicated sub-section entitled “Aims” has been included at the end of the Introduction. The section clarifies that the aims of this study are the identification of individual and contextual determinants that influence confidence in flu vaccination to create a strategic-organizational model encouraging the adherence through a nudge intervention.

To the authors best knowledge, there are no studies in the scientific literature about nudge interventions to implement vaccinations in university contexts.

Comment: The aim is the following: “identify individual and contextual determinants that influence confidence in flu vaccination in order to create a strategic-organizational model to encourage adherence through a nudge intervention”. My feeling is that the final discussion is limited to the first part of the aims (identify individual and contextual determinants). The second part, about the strategic-organizational model, is quite neglected. I would suggest either to elaborate more on this in the Conclusions or not to include this part in the aims.

Answer: The referee well focused the scopes of the conducted analysis. Aiming to clarify this aspect, the authors further elaborated the discussion. In particular, the revised text deepens on the way to define a strategic-organizational model to encourage the adherence through the implementation of a nudge intervention in the university context. This approach is based on the creation of a specific support structure with psychologists and medicals providing the interventions detailed in the manuscript.

Comment: Please specify what ASL stays for.

Answer: The authors are grateful to the referee for detecting this typo. The revised text specifies that the acronym ASL stands for Local Health Authority.

Reviewer 2 Report

The manuscript entitled "Individual and Contextual Determinants of Flu Vaccination Adherence: A University Nudge Intervention" reports a nudge intervention in the University of Salerno to promote vaccine adherence among employees of the academia, also individual and contextual determinants influencing adherence were identified. Some amendments are required for the manuscript:

- The information about the vaccine campaign could be provided in the M&M part instead of introduction.

- The frequency values in the tables should be revised. Some sum of the values are not 100% (e.g. gender, contact with flu etc).

- Always typing of the disease should be as "COVID-19".

- Table 4: It is better to write the SD values starting with zero as "0.xxx".

- Previous studies should be discussed more in Discussion.

Author Response

The authors are grateful for the careful reading and the precious comments received. The suggestions formulated by the reviewers have been highly supportive in the improvement of the manuscript and deepening of several key aspects.

Comment: The manuscript entitled "Individual and Contextual Determinants of Flu Vaccination Adherence: A University Nudge Intervention" reports a nudge intervention in the University of Salerno to promote vaccine adherence among employees of the academia, also individual and contextual determinants influencing adherence were identified. Some amendments are required for the manuscript:

- The information about the vaccine campaign could be provided in the M&M part instead of introduction.

- The frequency values in the tables should be revised. Some sum of the values are not 100% (e.g. gender, contact with flu etc).

- Always typing of the disease should be as "COVID-19".

- Table 4: It is better to write the SD values starting with zero as "0.xxx".

- Previous studies should be discussed more in Discussion.

Answer: The authors are grateful to the referee for his accurate reading and constructive comments.

The manuscript has been improved following the indications provided by the reviewer. The added/modified text has been highlighted in green.

The text has been reworked in some of its parts, so the authors find it useful to leave the information about the vaccination campaign in the introduction

The tables were modified according to the suggestions provided by the referee, in particular, the frequency values were corrected as well as SD values.

The wording “COVID-19” has been used throughout the manuscript, as suggested by the reviewer.

The authors agree with the referee about the usefulness of expanding the studies reviewed in the discussion section, and revised the manuscript accordingly.

Reviewer 3 Report

This is an interesting paper and will be of interest to researchers examining vaccine hesitancy and those policy makers investigating implementing vaccination campaigns

The main issue concern with the paper is with the claim that the nudge intervention made employees more responsible to protecting the health of the academic community and encouraged good adherence to the vaccination campaign. However, as acknowledged one of the limitations of the paper is that it did not include a control sample of those who did not become part of the campaign. Therefore, to say that the nudge intervention made employees more responsible seems unproven. This is only the case for those who became part of the campaign. Nonetheless, there are correlations between the variable measured and those willing to have a vaccination and the are interesting and worthy of publication. Therefore, it is suggested these should be the focus of the paper and the casual link be removed.

There are some other small areas to consider for improvement and these are listed below:

On page 4, line 150 it is stated that only Y1 of the STAI-Y inventory was used. While it is understandably why this is the case it might be useful to give a short explanation as to why only this part of the STAI-Y was used.

Page 4, line 183, the sentence states 43% of the sample were employed and then goes on to say 53.9% were professors etc. Are professors not also employed by the university. Can the difference between these two groups of university employees be given a little more explanation.

Page 7, Table 4, line 2 the word should read 'Employees' 

Otherwise this was a well written and well referenced paper.

Author Response

The authors are grateful for the careful reading and the precious comments received. The suggestions formulated by the reviewers have been highly supportive in the improvement of the manuscript and deepening of several key aspects.

Comment: The main issue concern with the paper is with the claim that the nudge intervention made employees more responsible to protecting the health of the academic community and encouraged good adherence to the vaccination campaign. However, as acknowledged one of the limitations of the paper is that it did not include a control sample of those who did not become part of the campaign. Therefore, to say that the nudge intervention made employees more responsible seems unproven. This is only the case for those who became part of the campaign. Nonetheless, there are correlations between the variable measured and those willing to have a vaccination and the are interesting and worthy of publication. Therefore, it is suggested these should be the focus of the paper and the casual link be removed.

Answer: The authors express their gratitude to the referee for his constructive comments The added/modified text has been highlighted in blue.

The authors reflected on the concept of “collective responsibility2, both in the discussions and in the conclusions, where they included future directions for research.

Comment: There are some other small areas to consider for improvement and these are listed below:

On page 4, line 150 it is stated that only Y1 of the STAI-Y inventory was used. While it is understandably why this is the case it might be useful to give a short explanation as to why only this part of the STAI-Y was used.

Answer: The authors are grateful to the referee for showing interest in this aspect.

The authors administered only the Y1 scale of the STAI-Y inventory, relating to state anxiety, because they were only interested in evaluating how the subject felt in the specific situation at the time the questionnaire was administered.

Comment: Page 4, line 183, the sentence states 43% of the sample were employed and then goes on to say 53.9% were professors etc. Are professors not also employed by the university. Can the difference between these two groups of university employees be given a little more explanation.

Page 7, Table 4, line 2 the word should read 'Employees'

Answer: The authors are grateful to the referee for detecting this typo.

The authors specify the role of employees into administrative employees and teachers.

Round 2

Reviewer 3 Report

Thank you for addressing the concerns raised in the original review. In my view the paper is now ready for publication.